# Endoscopic and Surgical Removal of Gastrointestinal Foreign Bodies in Dogs: An Analysis of 72 Cases

**DOI:** 10.3390/ani12111376

**Published:** 2022-05-27

**Authors:** Cristina Di Palma, Maria Pia Pasolini, Luigi Navas, Andrea Campanile, Francesco Lamagna, Gerardo Fatone, Fabiana Micieli, Ciro Esposito, Daniela Donnarumma, Valeria Uccello, Barbara Lamagna

**Affiliations:** 1Department of Veterinary Medicine and Animal Production, University of Naples “Federico II”, Via Federico Delpino 1, 80137 Naples, Italy; pasolini@unina.it (M.P.P.); luigi.navas@unina.it (L.N.); lamagna@unina.it (F.L.); fatone@unina.it (G.F.); fabiana.micieli@unina.it (F.M.); blamagna@unina.it (B.L.); 2Veterinary Practitioner, Endovet Professional Association, 82100 Benevento, Italy; andreacampanile@hotmail.it; 3Veterinary Practitioner, 80100 Naples, Italy; ciroesposito87@gmail.com (C.E.); danieladonnarumma@libero.it (D.D.); 4Veterinary Center ASL Napoli 1, 80145 Naples, Italy; dot.uccellovaleria@gmail.com

**Keywords:** foreign body, dog, gastrointestinal, endoscopy, surgery

## Abstract

**Simple Summary:**

The ingestion of foreign bodies is common in dogs. The aim of this multicentre retrospective study was to evaluate the most common location and type of gastrointestinal foreign bodies in dogs. Animal signalment, clinical factors and outcomes of dogs undergoing surgical or endoscopic treatment for foreign body removal were also examined. A high rate of affected dogs was young, male, and of a large breed. The main foreign body location was the stomach; endoscopic removal was associated with a high rate of success in gastric foreign bodies, whereas enterectomy and multiple surgical incisions were linked to a poor outcome.

**Abstract:**

In emergency veterinary practice, gastrointestinal foreign body (GFB) removal is a common procedure that is performed with different techniques, such as endoscopy or surgery. The aims of this retrospective, multicentre, clinical study were to report the common locations and types of objects recovered and to investigate clinical factors and outcomes in dogs after surgical or endoscopic treatment for GFB removal. Records of dogs with a GFB diagnosis referred to the Teaching Veterinary Hospital or treated in three different veterinary hospitals from September 2017 to September 2019 were examined. The data obtained from each case included breed, age, clinical signs at presentation, duration of clinical signs, type and location of the GFB, treatment, length of hospitalisation and outcome. Seventy-two dogs were enrolled in the study. There were 42 males (58%) and 30 females (42%). The median age was 36 months (range: 3 months to 8 years). Endoscopic retrieval was performed in 56% of GFBs (located in the stomach or duodenum), whereas 44% of dogs underwent surgery. The type of FB detected varied greatly: kid toy (14%), metallic object/coin (13%), cloth (13%), sock (8%), ball (8%), plastic material (8%), peach stone (7%), fishhook (6%), sewing needle (4%), hair tie (4%), pacifier (3%), plant materials (3%) and others (9%). Moreover, the FBs were classified as sharp (13%, *n* = 9), pointed (33%, *n* = 24), blunt (26%, *n* = 19), or linear (28%, *n* = 20). In this study, 68% of FBs were localised in the stomach, 25% in the intestinal tract (50% duodenum, 28% jejunum, and 22% ileum), and 7% in both the stomach and small intestine. The type of GFB was not significantly associated with age, site or breed. There was a significant association between the type of GFB and sex: if the dog was male, there was a 38% probability of ingesting linear GFBs. The dog survival rate was 100% in cases treated by gastric endoscopic or surgical removal, 94% in cases treated with enterotomy and 33% in cases in which enterectomy was necessary. Enterectomy and multiple surgical sites were associated with a poor outcome. The presence of vomiting for more than 24 h was significantly associated with death.

## 1. Introduction

In emergency practice, gastrointestinal foreign body (FB) ingestion is common in dogs. Foreign bodies can pass through the gastrointestinal tract without showing clinical signs, or they can cause damage and have a clinical impact, as in cases with partial or complete obstruction [1]. In human practice, the time of ingestion of FBs is known in most cases, and only 10–20% of ingested FBs require medical treatment [2], while others pass through the digestive system without any medical interventions. In veterinary medicine, the owner’s common lack of awareness regarding the time of FB ingestion results in diagnosis and treatment occurring in relation to the onset of clinical signs [3]; thus, the actual percentage of FB that passes through the gastrointestinal tract without requiring any treatment in dogs is unknown.

The nature of the FB, the degree of obstruction (complete or partial), and the presence of perforation have an influence on clinical presentation and treatment. The common clinical signs associated with gastrointestinal FB ingestion are vomiting, abdominal pain, anorexia, absence of defecation, and diarrhoea [1,3,4]. Vomiting is the most represented clinical sign in upper gastrointestinal obstruction, with a median duration of two or three days after ingestion before presentation [1,5]. Chronic vomiting may lead to hypovolemic shock that should be stabilised to improve the general condition before anaesthesia and to reduce gastrointestinal damage from poor intestinal perfusion. [6]. Furthermore, gastrointestinal FB obstructions are associated with an acid–base imbalance and alterations of electrolyte concentrations, such as hypochloremia and metabolic alkalosis [5].

Gastrointestinal FB removal is most frequently reported in young dogs belonging to medium or large breeds [1,3,4,7], and male dogs are represented more than females [8,9]. In the human and veterinary literature, the classification of GFBs can be based on their capability to cause partial or complete obstruction or on their nature. In particular, the shape of GFBs divides them into two large groups: linear and discrete GFBs. Furthermore, the nature of an FB (shape, composition, and sharpness of edges) and its location within the gastrointestinal tract influence its treatment (endoscopy or surgery) and outcome [8]. Upper gastrointestinal endoscopy is a minimally invasive treatment with a high success rate. In particular, endoscopic procedures are the first choice for removing FBs in the oesophageal and gastric regions of the gastrointestinal tract [9]. Surgical intervention is often necessary if a perforation is suspected or if the object has not progressed and FB endoscopic removal is not possible.

The aims of this retrospective, multicentre, clinical study were to report the most common locations and types of objects recovered and to investigate clinical factors and outcomes in dogs undergoing surgical or endoscopic treatment for GFB removal.

## 2. Materials and Methods

Medical records of dogs with a GFB diagnosis treated as first opinion practice in the Teaching Veterinary Hospital of the University of Naples “Federico II” or treated in three different veterinary hospitals of the urban area of Naples from September 2017 to September 2019 were examined. The study was conducted under approval record PG/2022/0049633 in 27 April 2022 by the OPBA (Organismo Preposto al Benessere Animale) of the University of Naples “Federico II”.

The study included dogs with FBs lodged in the gastrointestinal tract, ranging from the stomach to the small intestine. Data obtained from each dog included breed, age, body weight, clinical signs at presentation (vomiting, temperature, absence of defecation, and abdominal pain), duration of clinical signs (length of time from ingestion, if known, or length of time from the beginning of clinical signs to diagnosis), type and location of FB, treatment (endoscopy, gastrotomy, enterotomy or enterectomy), and duration of hospitalisation and outcome (survival, euthanasia, and death). In all cases, the diagnosis of clinical suspicion was supported by a diagnostic imaging examination (radiographic and/or ultrasound examination).

The foreign bodies were classified as pointed (stone, peach stone, or piece of wood), blunt (ball or latex teat), sharp (fishhook or needle), and linear (carpet, plastic bag, or cloth) according to the shape classifications made by Evans in 1994 and Chaves in 2004 [10,11].

A flexible videogastroscope (Storz outer diameter 7.8 mm, working channel diameter 2.8 mm, and working length 140 cm) was employed in the endoscopic procedure. Some different ancillary tools were used, such as Dormia baskets of different sizes, grasping forceps, rat tooth forceps, net baskets, and polypectomy snares. The choice of instruments depended on the endoscopist and the patient’s clinical status/condition (type and location of FB). The gastrointestinal surgeries were performed by four different expert surgeons. A ventral midline celiotomy was performed, and the surgeons evaluated the entire gastrointestinal tract via palpation and direct visualisation of the specific FBs. After FB identification, the surgery (gastrotomy, enterotomy, enterectomy, or multiple incisions) was performed. Termino-terminal anastomosis was performed during enterectomy. The perioperative treatment in the different clinics was similar, and the treatment protocols involved the administration of intravenous crystalloids (Ringer’s Lactate, Normosol-R and 0.9% saline solution), antibiotics (metronidazole, ampicillin, amoxicillin/clavulanic acid), gastrointestinal protectants and antiemetics (sucralfate, proton pump inhibitors, maropitant).

All numerical data were recorded using a computerised spreadsheet (Microsoft Excel 2016). Statistical analyses were performed using the Statistical Analysis System (SAS; SAS Institute, Cary, NC, USA). The results are expressed as the mean ± standard deviation (SD) for parametric data, whereas nonparametric data are reported as the median (minimum, maximum).

The chi-squared test (χ^2^) was used to compare differences in categorical variables, with *p* values lower than 0.05 indicating statistical significance (gender/FB class, age/FB class, and site/abdominal pain). When a low frequency of distribution of the values was found in the contingency table, a Fisher’s exact test (*p*-value < 0.05 considered significant) was used (FB class/site, FB class/weight, FB class/vomiting, FB class/absence of faeces, FB site/vomiting, FB site/absence of faeces, FB site/weight, FB site/vomiting, FB site/pain, follow up/age, follow up/weight, follow up/vomiting, follow up/pain, follow up/faeces, follow up/treatment).

## 3. Results

Animal and clinical records—72 dogs were enrolled in the study, including 42 dogs of 22 different pure breeds and 30 mixed-breed dogs. Twenty-two percent were small-sized dogs (<9 kg; *n* = 16), 18% were medium-sized dogs (9–20 kg; *n* = 13), 53% were large-sized dogs (20–40 kg; *n* = 38), and 7% were ultra large-sized dogs (>40 kg; *n* = 5). The mean ± SD body weight was 22.46 ± 12.17 kg, and the median body weight was 24.5 kg (range: 2–48 kg). There were 42 males (58%) and 30 females (42%). The median age was 3 years (range: 3 months to 8 years) (Table 1). Dogs aged ≤ 12 months accounted for 19% of the cases, 56% were aged from 12 to 48 months, and 25% were >48 months.

Vomiting was the most common clinical sign (82%); the time from initial presentation of vomiting to the diagnosis of a gastrointestinal FB was <12 h in 15%, 12–24 h in 54%, 24–36 h in 17%, and >36 h in 14% of cases. At the time of clinical presentation, 57% of dogs presented with abdominal pain, and 31% presented with absence of defecation.

Type and Location of GFBs—The type of FB detected varied greatly: kid toy (14%), metallic object/coin (13%), cloth (13%), sock (8%), ball (8%), plastic material (8%), peach stone (7%), fishhook (6%), sewing needle (4%), hair tie (4%), pacifier (3%), plant materials (3%) and others (9%) (Figure 1).

Moreover, the FBs were classified as sharp (13%, *n* = 9), pointed (33%, *n* = 24), blunt (26%, *n* = 19), or linear (28%, *n* = 20) (Figure 2).

In this study, 68% of FBs were localised in the stomach, 25% in the intestinal tract (50% duodenum, 28% jejunum, and 22% ileum), and 7% in both the stomach and small intestine.

Treatment and Outcome—Endoscopic retrieval was successfully performed in 56% (*n* = 40/72) of GFBs (localised in the stomach or duodenum) with 100% success, whereas 44% of dogs underwent surgery (*n* = 32/72; gastrotomy 46%, *n* = 13/72; enterotomy 42%; *n* = 15/72; enterectomy 3%, *n* = 1/72; gastrotomy/enterotomy 3%, *n* = 1/72; gastrotomy/enterectomy 3%, *n* = 1/72 and enterotomy/enterectomy 3%, *n* = 1/72). The length of hospitalisation was ≤ 1 day in 42% (*n* = 30/72, patients treated by endoscopy), 2–3 days in 55% (*n* = 40/72; *n* = 13/40, 33% endoscopy and *n* = 27/40, 67% surgery), and >3 days in 3% (*n* = 2/72; 50% endoscopy and 50% surgery) (Figure 3).

The survival rate was 100% in all cases of foreign bodies with exclusive gastric location treated by both endoscopic and surgical removal, 94% in cases treated with enterotomy and 33% in cases where enterectomy was necessary. Two of the three deceased dogs had undergone surgery with multiple incisions (enterectomy/enterotomy or gastrotomy/enterectomy) (Figure 4).

The type of GFB was not significantly associated with age, site or breed. The presence of vomiting for more than 24 h was significantly associated with death. There was a significant association between the type of GFB and sex. If the dog was male, there was a 38% probability of ingesting linear GFBs (OR = 4). The location of the GFB in the intestinal tract was significantly associated with abdominal pain and the absence of defecation (*p* < 0.001). No link was established between negative follow up (death) and body temperature or age. Unfavourable outcome in the follow up was associated with enterectomy and multiple gastrointestinal incisions (0.0035; *n* = 3/72).

## 4. Discussion

The results of the study indicated that dogs of large breeds (53%) were the most commonly affected, with a median age of three years, in agreement with the literature [1,5]. Furthermore, males ingested FBs more often than females. Capak et al. (2001) speculated that the preponderance of male dogs with FBs may be explained by the preference of owners for male dogs, particularly in the case of pure breeds. Breeders usually prefer female dogs that are mostly kept in kennels, which reduce the risk of ingesting FBs [4]. This hypothesis cannot explain our results relative to a sample in which mongrel dogs are overrepresented. FB ingestion was associated with behavioural disorders, such as anxiety, attachment, and hypersensitivity–hyperactivity syndrome [12], which occur more commonly in male than female dogs, according to recent studies [13,14]. In the study by Masson et al. [12], no effect of sex on FB ingestion was established. In the medical records that we examined, there were no data related to the presence/absence of behavioural problems. Additional studies are required to establish the effect of sex on these behavioural disorders. According to Bradshaw et al. [15], solitary play with nonedible objects appeared to be derived from predatory behaviour that is exhibited more in male than in female animals [16], and during this solitary play, dogs preferred toys to be altered and dismembered, such as tissues [15]. Furthermore, according to Masson et al. (2021), the type of ingested FB was not linked to a specific behavioural disorder but rather to an individual preference of a dog to ingest the same type of nonedible object [12]. Regardless, behavioural pathology could underlie both the ingestion of FBs and shredding of nonedible objects. Moreover, the young age of the patients affected could be explained by the presence of behavioural disorders, such as hypersensitivity–hyperactivity syndrome, which is a typical neurological disorder of children in human medicine [14]. Thus, a behavioural consultation should always be required in these patients to evaluate underlying behavioural disorders and management mistakes [12].

In the present study, the main FB location was in the stomach (68%), followed by the duodenum (11%); therefore, the upper gastrointestinal tract was mostly affected. These data agreed with those of other studies, such as Boag et al. (50% in the stomach) [5] and Poggiani et al. (78% in the stomach) [9]. In contrast, Hayes and other researchers recorded 16% of FBs in the stomach, while the jejunum was the most common location [1]. These different locations could be explained by an early presentation by dog owners, as suggested in other studies [1].

Furthermore, the results suggested that endoscopic and surgical procedures recorded high rates of success in the management of FB in the upper gastrointestinal tract; endoscopy is especially associated with a short hospitalisation time, and it can be considered the procedure of choice for the removal of gastric FBs due to its low invasiveness and shorter performance time compared to surgical procedures [17]. A poor prognosis seems to be correlated with the type of surgery, multiple surgical incisions and enterectomy, in agreement with data in the literature [8,18]. The survival rate of dogs undergoing enterectomy reported in our study (33%; *n* = 1/3) appears low, and these data could be justified by the extremely small number of enterectomies performed. Perhaps, the attention of the owners and the diagnostic imaging opportunities offered by veterinary clinics can lead to early interventions after the ingestion of a foreign body, preventing serious alterations to the affected enteric tract. A longer duration of clinical signs and partial versus complete obstruction influence this outcome. Additionally, gastrointestinal obstruction is associated with disturbances of fluid balance, acid–base imbalance and electrolytes, which can also greatly affect the outcomes. Unfortunately, there was no information in our study regarding the alterations found in subjects at the time of diagnosis or about the complications that led to the death of the patients. Consequently, it was not possible for us to evaluate the influence of these conditions on the outcome. In the study by Hayes, a high rate of dogs died postoperatively, and those patients presented signs of peritonitis linked to enteric wound breakdown [1]. According to Lopez et al. (2021), dogs undergoing intestinal resection and anastomosis were associated with a higher risk of intestinal dehiscence than dogs undergoing enterotomy [18]. It is possible that a higher rate of dehiscence could explain our results in patients undergoing enterectomy, but data concerning postoperative complications were missing in our study. No association between poor prognosis and age was present, although in another study, older patients had a higher risk of intestinal dehiscence and a negative prognosis due to delayed healing; more research is needed to assess the link between age and intestinal wounds in dogs [18].

The presence of vomiting for more than 24 h was significantly associated with a poor prognosis; perhaps, the hypovolemic condition after persistent vomiting could cause a decrease in intestinal perfusion, compromising the integrity of the wall with an increasing risk of septic peritonitis. Medical records about perfusion parameters were missing in this study, but Hoffman et al. (2021) found that dogs with persistent vomiting had an increased likelihood of receiving an enterectomy [6].

There are many limitations of a multicentre retrospective study. Some data about the clinical presentation, haematobiochemical parameters, medical therapy, and complications associated with poor outcomes were missing. The choice of treatment may have been influenced by the clinician’s experience and the requests of dog owners. Furthermore, there was no information about the presence of behavioural disorders in the affected dogs.

Emergency veterinarians should consider the possibility of a behavioural disorder in the case of a dog with a gastrointestinal foreign body, and they should suggest a behavioural specialist assessment for these dogs. In addition, more data should be collected in order to create intervention guidelines based on the type and size of the ingested foreign body, as in human medicine [19]. Future studies with a higher number of dogs could provide useful data to create guidelines in emergency veterinary practice.

## 5. Conclusions

In this study, a high rate of large dog breeds of a young age ingested foreign bodies, and the principal location was the stomach. Other studies are required to establish gender predisposition and the association of foreign body ingestion with the presence of behavioural disorders. Endoscopic gastric foreign body removal was generally associated with a high success rate, while persistent vomiting, enterectomy and multiple surgical sites were linked to a poor postoperative outcome. According to the authors, a behavioural investigation should be performed in dogs who have ingested a nonedible foreign body in order to prevent recurrence.

## Figures and Tables

**Figure 1 animals-12-01376-f001:**
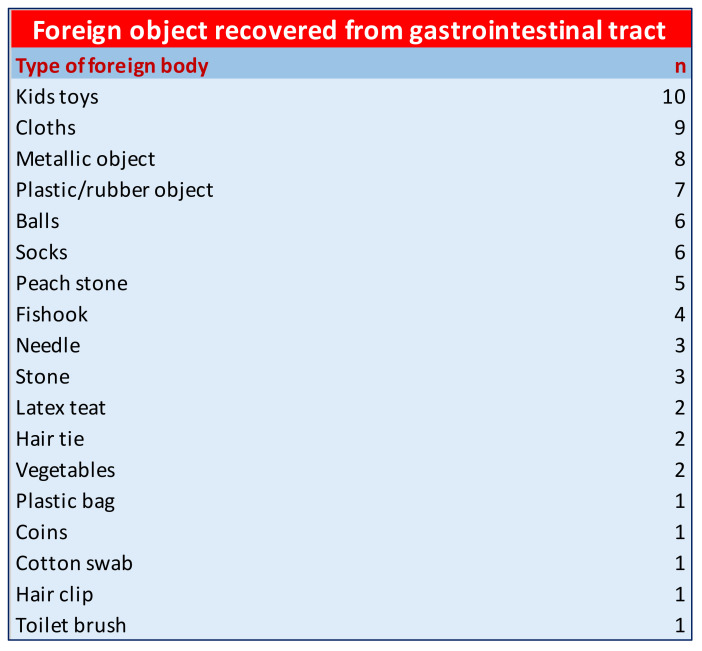
Data about the type of foreign bodies detected.

**Figure 2 animals-12-01376-f002:**
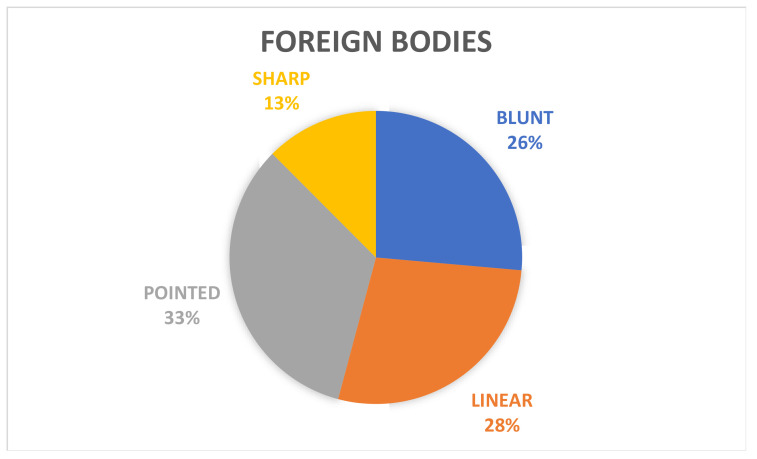
Graph showing the percentage of the different types of foreign bodies recovered from gastrointestinal tract.

**Figure 3 animals-12-01376-f003:**
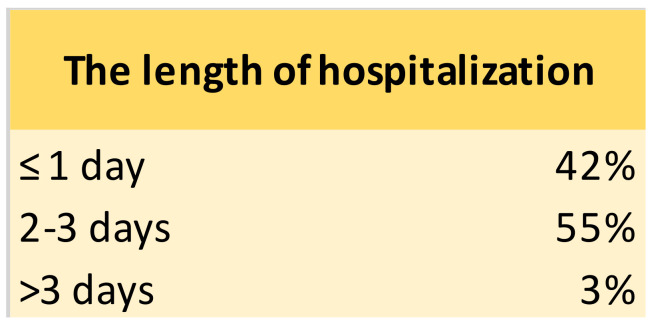
Results about the length of hospitalization.

**Figure 4 animals-12-01376-f004:**
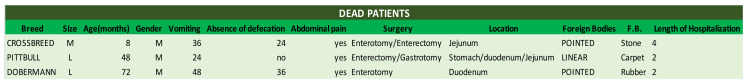
Data about the three dogs that did not survive.

**Table 1 animals-12-01376-t001:** Summary table of patients’ details at presentation.

Table of Patients Details at Presentation	
**Gender** ( male/female)	42/30
**Age** (years)	
median (range)	3 (0.25–8 )
**Weight** (kg)	
mean (sd)	22.46 (12.17)
**Breed**	
Small-sized dogs <9 kg	16
Medium-sized dogs 9–20 kg	13
Large-sized dogs 20–40 kg	38
Ultra large-sized dogs >40 kg	5

## Data Availability

The data are available upon request from the submitting author.

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
