# Peer review of "Endoscopic and Surgical Removal of Gastrointestinal Foreign Bodies in Dogs: An Analysis of 72 Cases"

_animals, 2022, doi:10.3390/ani12111376_

Round 1

Reviewer 1 Report

Summary: The aims of the study were retrospectively to analyse clinical cases related to the ingestion of foreign bodies in four different veterinary clinics over a 2-year period. Type of foreign body, gastrointestinal location and outcomes between dogs undergoing surgical versus endoscopic treatment were evaluated. 72 cases were included. The study showed that the stomach was the most common location, followed by the duodenum; that a high variety of objects can be found and finally, that endoscopic was associated with a high rate of success, whereas enterectomies were associated with a higher risk of death. In addition, it was found that the presence of vomiting for more than 24 hours was significantly associated with a poor prognosis.

General  comments 

Thank you for conducting this study. Although similar studies have been recently published, the study provides important epidemiologic information about the location and types of foreign bodies, as well demographic information and outcomes that are useful.

However, I consider that the paper would benefit from the addition and/or clarification of some aspects in the results and discussion. Please find below my comments:

The paper says that there was a significant association between the type of foreign body and sex, and if the dog was male, there was a 38% of probability of ingesting a linear foreign body. In the discussion, one of the potential reasons you mentioned was related to behavioural issues. Is there a relationship between behavioural issues and a preference for linear foreign bodies? That part of the discussion needs to be clarified.

It is interesting that you mention that only 33% of the cases that required enterectomy survived. That is a very low percentage compared to previous studies. It is not clear the number of dogs that underwent enterectomy, but if the numbers are low, that could be a contributing factor. You have reported only three deaths.

A longer duration of clinical signs and partial versus complete obstruction influence outcome. Also, gastrointestinal obstruction is associated with disturbances of fluid balance, acid-base and electrolytes, which also can greatly affect the outcomes. That info is not mentioned in the paper.  There was also no info about medical treatment and previous treatments. 

Cases were treated in four different veterinary clinics, including one teaching veterinary hospital.  Although the involvement of multiple institutions, can have the advantage of providing a higher number of clinical cases for analysis, different case management strategies, availability of diagnostic methods, surgeon availability, costs, and different levels of expertise among clinics can influence the results. Were all the cases first-opinion or referrals? There is no mention of the diagnostic methods used to confirm the presence of a GFB.

Specific comments

Line 49: there is not a clear connection between this sentence (human literature) and the rest of the paragraph.

Line 81: How many cases were diagnosed in the veterinary teaching hospital?

Line 87: time of ingestion and start of clinical signs can be very variable and will depend on many factors. For example, moderate and severe clinical signs can be more easily detected than subtle signs. It is reported that owners aware of foreign body ingestion usually present their animals significantly earlier than those not aware of foreign body ingestion, and that can also influence the outcome. What do you mean exactly with the length of time from ingestion and length of time from the beginning of clinical signs? Is this in regard to the time of surgical treatment?  to time of diagnosis? Is there a parameter for early versus delayed treatment? Prompt presentation, diagnosis and surgical intervention improve the outcome of gastrointestinal obstruction by foreign bodies.

Was there a difference in the duration of the endoscopies and the surgical treatments? That could be another contributing factor to the success rate.

Line 100:  certified surgeons?

Line 120-131: A summary table could be very useful.

Line 130: what about physical examination findings?  How many foreign bodies could be detected during abdominal palpation? 

Line 145 to 150: Please provide numbers in addition to the percentages.

Line 153: 100% survival was for endoscopic or surgical cases? The way that is written is confusing.

Line 156: Table of dead patients. Please specify that the age is given in months. Length of hospitalization has a spelling mistake. Here, it is very important to make the distinction between the length of clinical signs versus the length of hospitalization. A longer duration of clinical signs can be associated with poorer outcomes and longer hospitalization days can be caused by more complications present. Here, dog number 3 had only less 1d. Was this dog severely affected?

Although, you mention in the paper that clinicopathological findings were not available for all patients. If possible, data from these patients should be included to have a better context.

Line 161: what was the location?

Line 187: and is there a connection with the ingestion of foreign bodies in children with this syndrome? Are there reports of ingestion of foreign bodies associated with behavioural issues in dogs?

Line 189: 68% of the foreign bodies were found in the stomach and you discussed that one of the reasons was the early presentation. What were your criteria for early versus late presentation? How many cases were early presented versus late? Maybe early presentation contributed that the foreign body didn’t have time to move to the small intestine and caused more severe clinical signs. Thus, endoscopy had a better outcome compared to surgical procedures. 

Line. 203: that is a little bit unusual, especially if the patients were hospitalized at the time of death. Was necropsy performed? Or clinicopathological reports or reports of clinical signs suggestive of potential complications? Surgical reports that mention anything about the aspect of the intestine?

Line 224: What are your suggestions for future studies?

Reviewer 2 Report

Dear Authors,

there are minor corrections in the text that should be made:

  • Sentence in line 48: I would suggest to use the following formulation "and have a clinical impact" instead of "create a clinical impact".
  • Sentence in line 49: correction to the symbol - instead of "10+20%" it should be "10-20%".
  • Sentence in line 114: correction of the symbol for the P value.
  • Sentence in lines 121-123 (Results section): I would to change the classification of dogs according to body size; generally, breeds are classified as miniature and toy breeds (<5 kg), small breeds (5-10 kg), medium breeds (10-25 kg), large breeds (25-45 kg) and giant breeds (>45 kg). It is strongly recommended to use a standardised way of classifying breeds according to body size.
  • Sentence in line 179: instead of just writing "Bradshaw's study" reference should be cited correctly. The same applies for the reference cited in lines 193 and 204.
  • Sentence in line 191: I would recommend to use upper gastrointestinal tract instead of high.

In the "Results" section the text should be linked to the figures: i. e. when writing about the types of the foreign bodies (sentences in lines 132-135) it should be stated in the same paragraph to which figure the data presented pertain.
